# Distinct Plasma Concentrations of Acyl-CoA-Binding Protein (ACBP) in HIV Progressors and Elite Controllers

**DOI:** 10.3390/v14030453

**Published:** 2022-02-23

**Authors:** Stéphane Isnard, Léna Royston, John Lin, Brandon Fombuena, Simeng Bu, Sanket Kant, Tsoarello Mabanga, Carolina Berini, Mohamed El-Far, Madeleine Durand, Cécile L. Tremblay, Nicole F. Bernard, Guido Kroemer, Jean-Pierre Routy

**Affiliations:** 1Infectious Disease and Immunity in Global Health Program, Research Institute of McGill University Health Centre, Montreal, QC H4A 4J1, Canada; lena.royston@mail.mcgill.ca (L.R.); john.lin@mail.mcgill.ca (J.L.); brandon.fombuena@mail.mcgill.ca (B.F.); simeng.bu@mail.mcgill.ca (S.B.); sanket.kant@mail.mcgill.ca (S.K.); tsoarello.mabanga@mail.mcgill.ca (T.M.); carolina.berini@muhc.mcgill.ca (C.B.); nicole.bernard@mcgill.ca (N.F.B.); 2Chronic Viral Illness Service, McGill University Health Centre, Montreal, QC H4A 4J1, Canada; 3CIHR Canadian HIV Trials Network, Vancouver, BC V6Z 1Y6, Canada; 4Division of Infectious Diseases, Geneva University Hospitals, 1205 Geneva, Switzerland; 5Division of Experimental Medicine, McGill University, Montreal, QC H4A 3J1, Canada; 6Instituto de Investigaciones Biomédicas en Retrovirus y SIDA (INBIRS), CONICET-Universidad de Buenos Aires, Buenos Aires C1121ABG, Argentina; 7Centre de Recherche du Centre Hospitalier de l’Université de Montréal, Montréal, QC H2X 0A9, Canada; mohamed.el.far.chum@ssss.gouv.qc.ca (M.E.-F.); madeleine.durand@gmail.com (M.D.); c.tremblay@umontreal.ca (C.L.T.); 8Département de Microbiologie, Immunologie et Infectiologie, Université de Montréal, Montreal, QC H3T 1J4, Canada; 9Division of Clinical Immunology, McGill University Health Centre, Montreal, QC H4A 3J1, Canada; 10Centre de Recherche des Cordeliers, Equipe labellisée par la Ligue Contre le Cancer, Université de Paris, Sorbonne Université, Inserm U1138, Institut Universitaire de France, 75006 Paris, France; kroemer@orange.fr; 11Metabolomics and Cell Biology Platforms, Institut Gustave Roussy, 94805 Villejuif, France; 12Department of Biology, Institut du Cancer Paris CARPEM, Hôpital Européen Georges Pompidou, AP-HP, 75015 Paris, France; 13Division of Hematology, McGill University Health Centre, Montreal, QC H4A 3J1, Canada

**Keywords:** elite controllers, HIV, acyl-coA-binding protein, autophagy

## Abstract

HIV elite controllers (ECs) are characterized by the spontaneous control of viral replication, and by metabolic and autophagic profiles which favor anti-HIV CD4 and CD8 T-cell responses. Extracellular acyl coenzyme A binding protein (ACBP) acts as a feedback inhibitor of autophagy. Herein, we assessed the circulating ACBP levels in ECs, compared to people living with HIV (PLWH) receiving antiretroviral therapy (ART) or not. We found lower ACBP levels in ECs compared to ART-naïve or ART-treated PLWH (*p* < 0.01 for both comparisons), independently of age and sex. ACBP levels were similar in ECs and HIV-uninfected controls. The expression of the protective HLA alleles HLA-B*27, *57, or *58 did not influence ACBP levels in ECs. ACBP levels were not associated with CD4 or CD8 T-cell counts, CD4 loss over time, inflammatory cytokines, or anti-CMV IgG titers in ECs. In ART-treated PLWH, ACBP levels were correlated with interleukin (IL)-1β levels, but not with other inflammatory cytokines such as IL-6, IL-8, IL-32, or TNF-α. In conclusion, ECs are characterized by low ACBP plasma levels compared to ART-naïve or ART-treated PLWH. As autophagy is key to anti-HIV CD4 and CD8 T-cell responses, the ACBP pathway constitutes an interesting target in HIV cure strategies.

## 1. Introduction

The recent description of the Esperanza patient who achieved a possible spontaneous HIV sterilizing cure represents a beacon of hope for people living with HIV (PLWH) [1,2,3,4].

Elite controllers (ECs) can control HIV replication without antiretroviral therapy (ART) for a significant period of time. Studies of ECs led to a critical understanding of the factors associated with HIV control, offering possible clues for a HIV cure [3,5].

EC status has been shown to rely on a complex set of genetic and immunometabolic characteristics [6,7]. Protective human leukocyte antigen (HLA) alleles have been associated with robust anti-HIV CD4 and CD8 T-cell responses [8,9,10,11]. Moreover, metabolic pathways in immune cells are emerging as key factors for viral control in ECs [12,13,14]. Notably, our group has shown that the metabolic inputs of autophagy and lipophagy contribute to the HIV-specific CD4 and CD8 T-cell response in ECs [15,16]. Angin et al. first showed that HIV-specific CD8 T-cells display a restrictive glucose dependency in the majority of PLWH, which is a distinct metabolic defect not observed in ECs [17]. We confirmed that the superior CD8 T-cell responses observed in ECs are linked to their ability to use lipid and glutamine as metabolic inputs during autophagic processes [15,16]. Autophagy is involved in cellular homeostasis and cytoprotection, with an anti-aging effect. Autophagy involves master regulators of cellular metabolism, such as the mammalian target of rapamycin (mTOR) and the 5′ AMP-activated protein kinase (AMPK), combined with the selective removal of protein aggregates and organelles such as mitochondria, the endoplasmic reticulum, peroxisomes, lysosomes, and lipid droplets [18]. Autophagy and lipophagy, a form of selective autophagy, are regulated by different factors including acyl-coA-binding protein (ACBP). This protein, also known as diazepam binding inhibitor (DBI), is involved in intracellular bioenergetic reactions and serves as an extracellular feedback inhibitor of autophagy [19,20,21]. ACBP is conserved in eukaryotes and is expressed by all nucleated cells, underscoring its importance as a cellular metabolic hub. Within cells, ACBP binds to activated, acyl-coA-bound medium-chain fatty acids and shuttles them between cellular organelles for energy production. While intracellular ACBP has been shown to promote autophagy, lipophagy, and oxidative phosphorylation [22,23], secreted extracellular ACBP inhibits autophagy and stimulates appetite to increase nutrient uptake [19,20,21,24]. Hence, ACBP can be considered a metabolic and neuroendocrine factor that regulates bioenergetic and cellular functions in a context-dependent manner [21,24].

Given the influence of ACBP on cellular metabolism, and the distinctive immunometabolic features of ECs, we measured circulating ACBP in ECs, ART-naïve, and ART-treated PLWH. We found that ECs are characterized by low plasma ACBP concentrations.

## 2. Materials and Methods

### 2.1. Study Design

Blood samples were selected from PLWH who participated in the Canadian cohort of HIV-infected slow progressors (CIHR/CTN 247), and the HIV pathogenesis biobank [25]. From this cohort, we identified 37 ECs who maintained a HIV plasma viral load (VL) below 50 copies/mL in the absence of ART for at least 6 months, with CD4 T-cell counts greater than 200 cells/μL. Samples from ECs were collected between 2006 and 2020.

Samples from 27 ART-naïve participants in the chronic phase of the infection (at least 6 months after the estimated date of HIV acquisition) were obtained from the Montréal primary HIV infection cohort, the chronic viral illness service (CVIS) biobank, and the HIV pathogenesis biobank [26]. In addition, samples from 55 adult ART-treated PLWH were included from the Canadian HIV and aging cohort study, and the HIV pathogenesis biobank [27]. All participants had a VL below 50 copies/mL of plasma, and CD4 T-cell counts above 200 cells/μL. Samples from 31 adult HIV-uninfected donors from the Canadian HIV and aging cohort study, the CVIS biobank, and the HIV pathogenesis biobank were included as controls. Samples from ART-naïve, ART-treated, and control participants were collected during 2003–2012, 2011–2016, and 2002–2018 periods, respectively.

### 2.2. Laboratory Measurements

Blood samples were collected from participants and plasma was stored at −80°C until use, as previously described [25]. The quantification of plasma HIV VL was carried out using the RealTime HIV-1 assay (Abbott Laboratories, Chicago, IL, USA). Absolute CD4 and CD8 T-cell counts were measured by clinical labs using flow cytometry. HLA typing was performed by Sanger sequencing or next-generation sequencing using kits from GenDx (Utrecht, The Netherlands) [28].

ACBP plasma levels were measured for all samples in November 2021, using a human ACBP ELISA kit (Abnova Taiwan corporation, Taiwan) according to the supplier’s instructions. The markers of inflammation interleukin (IL)-1β, IL6, IL-8, and tumor necrosis factor alpha (TNF-α) were quantified in plasma using the Meso Scale Discovery U-Plex Pro-Inflammatory Combo 4 kit (Meso Scale Discovery, Rockville, MD, USA). CMV IgG titers and total IL-32 were measured by ELISA as previously described [25,29,30]. All measurements were performed in duplicate.

### 2.3. Calculation of CD4 T-Cell Count Change

For ECs enrolled in the CIHR/CTN 247 study, sequential CD4 T-cell counts were prospectively combined with historical data from medical charts for each study participant. The rate of CD4 T-cell count change over time was computed using a linear regression analysis. The results were reported as annual changes in CD4 T-cell counts. The annual slope of CD4 T-cell count change was assessed for significant difference from “0” [29].

### 2.4. Statistical Analyses

GraphPad Prism 9.3.0 (GraphPad, La Jolla, CA, USA) and SPSS 24.0 (IBM SPSS, Chicago, IL, USA) were used for statistical analyses. The statistical significance of differences between groups was assessed using nonparametric Kruskal–Wallis’ tests with Dunn’s post-tests. A Spearman’s rank test was used to assess correlations. An α level of 5% was considered statistically significant (*p*-value). Multivariable linear regression analyses were performed using SPSS 24.0.

### 2.5. Ethical Considerations

Ethical approval was obtained from the McGill University Health Centre Research Ethics Board (REB), as well as from all the REBs of the institutions participating and recruiting individuals included in this study. All study participants provided written informed consent.

## 3. Results

### 3.1. Study Population

A total of 150 participants were included in this study. The ECs were younger than the ART-treated PLWH (45.5 vs. 54-year-old, *p* = 0.004), and exhibited higher CD4 T-cell counts (640, 310, and 546, *p* < 0.001 and 0.007 respectively) and higher CD4/CD8 ratios (1.0, 0.39, and 0.74, *p* < 0.001 and 0.006 respectively) than the ART-naïve and ART-treated PLWH. A similar percentage of female and male participants was observed between the groups (23, 23, 11, and 26 percent females in the ECs, ART-treated, ART-naïve, and control participants, respectively) (Appendix A). The EC participants remained aviremic for a median of 6 years (0.7–27).

The ART-naïve PLWH had a detectable VL (median 4.7 log_10_ copies/mL), whereas all the ART-treated PLWH and ECs had a VL below 50 copies/mL (1.6 log_10_ copies/mL). The ART-treated participants received treatment for a median of 13.7 years. In the EC group, 17 out of 37 participants harbored protective HLA-B*27, *B57, or *B58 alleles (Appendix A).

### 3.2. Circulating ACBP Levels Were Lower in ECs Compared to ART-Treated PLWH

Among PLWH, ACBP levels were lower in the ECs compared to the ART-naïve or ART-treated PLWH, with medians of 109.8, 238.7, and 264.6 ng/mL, respectively (*p* < 0.0001 for both comparisons) (Figure 1). Multivariable analyses showed that differences in age and sex could not explain these differences in ACBP levels. ACBP levels were similar in the ECs and uninfected controls (109.8 vs. 121.3 ng/mL, *p* = 0.33) (Figure 1). The ECs receiving statins or benzodiazepines had similar levels to those who did not take such medications (2 ECs were prescribed statins: ACBP levels 109.8 and 43.8 ng/mL; 1 EC was prescribed benzodiazepines: ACBP level 43.8 ng/mL). Three and seven participants, respectively, from the ART-treated and uninfected groups had particularly high levels of ACBP. However, their clinical characteristics such as diabetes status, weight, BMI, or age did not differ from the rest of the groups they belonged to (Appendix A). In the ECs, the duration of infection and follow-up were not associated with ACBP levels (r = −0.03, *p* = 0.88, Appendix A), and the ECs controlling infection for less than 5, 5 to 10, or more than 10 years had similar ACBP levels (Appendix A)

### 3.3. Markers of HIV Disease Progression Were Not Associated with ACBP Levels in ECs

ACBP levels were not associated with age in the ECs, ART-naïve, or ART-treated PLWH (r = −0.13, *p* = 0.44; r = −0.014, *p* = 0.94; r = 0.12, *p* = 0.37, respectively), whereas they were correlated with age in the uninfected controls (r = 0.46, *p* < 0.01). ACBP levels were not associated with CD4 T-cell counts, CD8 T-cell counts, or CD4/CD8 ratios, which are considered markers of HIV disease progression and risk factors for non-AIDS comorbidities [31,32,33,34]. However, ACBP levels were negatively correlated with CD4 and CD8 T-cell counts in the ART-naïve PLWH (Table 1).

The ECs harboring protective HLA alleles had similar ACBP levels to those who did not (80 vs. 117 ng/mL, *p* = 0.22, Appendix A and Appendix A).

The ECs with slopes of CD4 count change significantly below “0” had similar ACBP levels to those with stable or increasing CD4 T-cell counts (109.8 vs. 110.1 ng/mL, *p* = 0.89, Appendix A). Moreover, the ACBP level of the ECs was not associated with the slope of CD4 T-cell count change (correlation with annual CD4 slope r = 0.32, *p* = 0.07, Table 1). We previously found that the annual rate of CD4 T-cell count change was associated with CMV co-infection and anti-CMV IgG titers [29]. We did not find an association between ACBP levels and anti-CMV IgG titers in any group (Table 1). Moreover, we did not find any correlation between ACBP and IL-32 levels, a biomarker for control failure [25], in the ECs or ART-treated PLWH (r = 0.38, *p* = 0.13, and r = 0.03, *p* = 0.88, respectively).

Although ACBP levels have been linked with weight and BMI in obese people, we did not find any such association in our study groups (Table 1) [24].

In the ART-treated PLWH, we observed a significant positive correlation between ACBP and IL-1β concentrations (r = 0.29, *p* = 0.03, Appendix A), but not with IL-6, IL-8, or TNF-α levels (Appendix A). Such an association was not observed in the other groups.

## 4. Discussion

Our study demonstrates that ECs have significantly lower plasma ACBP levels compared to ART-naïve and ART-treated PLWH. These findings are consistent with the differences observed in T-cell metabolic pathways between ECs and other PLWH [15,16]. Lower plasma ACBP levels have been hypothesized in ECs, as extracellular ACBP inhibits autophagy and anti-HIV T-cell responses are favored by autophagy [15,16,24]. We observed a tendency of ECs with protective HLA to have lower ACBP levels, further suggesting that extracellular ACBP is linked with wearker anti-HIV responses.

ACBP levels in ECs were not influenced by participant characteristics such as age, sex, weight, or BMI. Furthermore, the HIV VL, treatment status and CD4 T-cell count of PLWH did not correlate with ACBP levels. ART-naive and ART-treated PLWH had similar levels of ACBP, suggesting that ART has no impact on ACBP levels [35]. However, both ART-naïve and ART-treated PLWH had higher ACBP plasma levels than ECs, suggesting that low ACBP is a patient-intrinsic, treatment-independent characteristic of ECs.

ECs have been shown to present with lower levels of inflammation compared to ART-naive and ART-treated PLWH [12,36,37]. However, other groups found elevated levels of the inflammatory markers IP-10 and TNF-α in ECs compared to controls, and those levels were not different from those of ART-treated PLWH [38]. In the group of ART-treated PWLH, we found an association between the inflammatory cytokine IL-1β and ACBP. This suggests that specific pro-inflammatory pathways could be responsible for ACBP induction in PLWH. Indeed, circulating ACBP levels were elevated in patients undergoing gut surgery, correlated with clinical signs of inflammation and high TNF-α levels [39]. Moreover, ACBP levels are elevated in the cerebrospinal fluid and the plasma of patients with Alzheimer’s disease and other neuroinflammatory conditions [40,41].

ACBP has been shown to promote food intake in animal models, and ACBP levels were higher in obese people [22,23,24]. Although weight gain is an issue for ART-treated PLWH, we did not observe any significant correlation between ACBP and either weight or BMI in our study, which may be underpowered for this kind of analysis.

Due to the small EC sample size, we were not able to optimally match participants and controls for age and sex. We founds that the usage of lipid-lowering agents or benzodiazepines did not influence ACBP levels in ECs. For the other groups, we did not have information on their treatment outside of antiretroviral drugs. Moreover, ACBP levels were only analyzed in plasma samples. The tissue expression of ACBP, notably in the liver and adipose tissue, should be compared in ECs vs. ART-naïve and ART-treated PLWH [42]. The comparatively low ACBP plasma levels of ECs could reflect a reduced secretion of ACBP, as well as a lower expression of intracellular ACBP. The cellular source and the mechanisms underlying the production of ACBP in ART-naïve and ART-treated PLWH, compared to ECs and uninfected controls, should be explored in future studies.

## 5. Conclusions

In summary, we detected lower ACBP plasma concentrations in ECs compared to ART-naïve or ART-treated PLWH. Low plasma ACBP levels could be used as a marker of ECs, reflecting this population’s particularly efficient immunometabolism. It would be interesting to stimulate autophagy by neutralizing the extracellular ACBP in ART-treated PLWH [24], and to investigate whether such a manipulation could facilitate the control of viral infections in suitable preclinical models. Exploring the ACBP pathway could also unravel new therapeutic targets to promote anti-HIV CD4 and CD8 T-cell responses in PLWH.

## Figures and Tables

**Figure 1 viruses-14-00453-f001:**
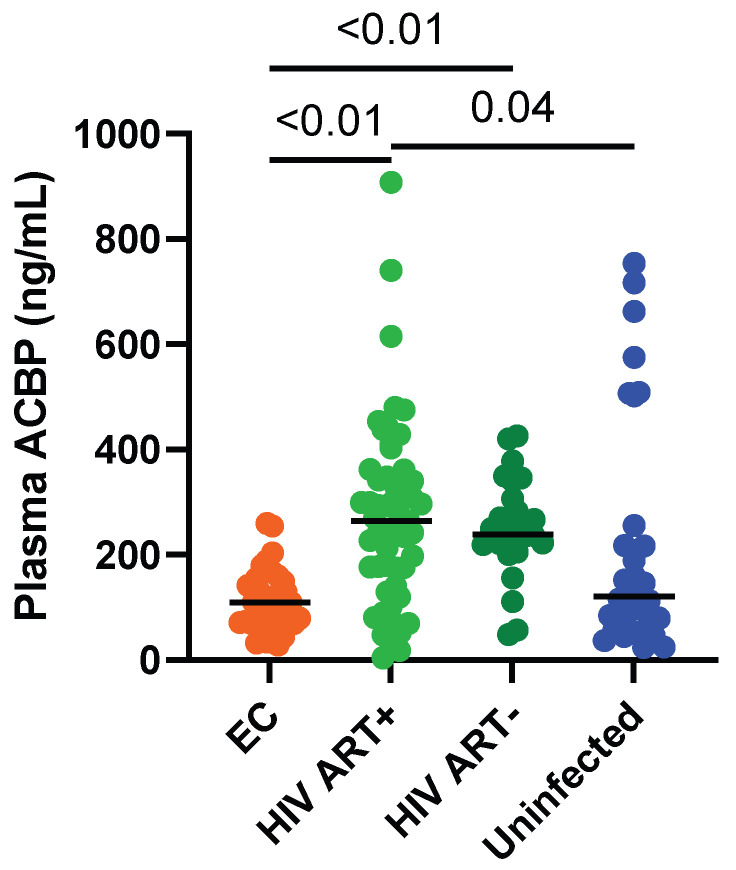
Plasma ACBP levels are lower in ECs compared to ART-naïve and ART-treated PLWH. Statistical significance measured via Kruskal–Wallis’s test. Black bars represent medians.

**Table 1 viruses-14-00453-t001:** Correlations between ACBP levels and HIV-disease progression markers.

ACBP vs.	EC		HIV ART+		HIV ART-		Uninfected Controls	
	**r**	** *p* **	**r**	** *p* **	**r**	** *p* **	**r**	** *p* **
CD4 count	−0.08	0.62	−0.06	0.64	**−0.48**	**0.01**	0.33	0.22
CD8 count	−0.14	0.41	−0.06	0.68	**−0.38**	**0.048**	**−0.53**	**0.04**
CD4/CD8 ratio	0.21	0.21	0.03	0.80	−0.12	0.56	**0.68**	**<0.01**
CD4 decay	0.32	0.07	ND		ND		ND	
Weight	−0.25	0.24	−0.002	0.99	0.14	0.69	ND	
BMI	−0.03	0.90	0.11	0.44	0.23	0.50	ND	
CMV IgG titers	−0.19	0.31	0.05	0.79	−0.06	0.8	0.12	0.66
Presence of protective HLA	0.2 *		ND		ND		ND	

* Mann–Whitney’s test between participants with vs. without protective HLA-B27, -B57, or -B58. ND: not done. Significant correlations are indicated in bold. ACBP: acyl-coA binding protein; BMI: body mass index; CMV: cytomegalovirus; Ig: immunoglobulin; HLA: human leukocyte antigen. Correlations performed with Spearman’s test.

## Data Availability

Data are available upon fair requisition to Jean-Pierre Routy at jean-pierre.routy@mcgill.ca.

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
