# Peer review of "Distinct Plasma Concentrations of Acyl-CoA-Binding Protein (ACBP) in HIV Progressors and Elite Controllers"

_viruses, 2022, doi:10.3390/v14030453_

Round 1
Reviewer 1 Report
This study from Isnard et al aims at assessing acyl-coA binding protein (ACBP) plasma levels in HIV elite controllers (EC) as compared with HIV progressors either treated by antiretroviral theray or ART naïve patients. Together with genetic and immunologic characteristics, metabolic pathways in immune cells are emerging as key factors for viral control. The group of JP Routy previously showed that the metabolic input of autophagy contributes to HIV specific T-cell responses in EC. As ACBP modulates autophagy, they performed a cross sectional study to measure circulating ACBP levels in EC (n= 37), ART naive HIV-infected patients (N= 27) and ART-treated HIV+ patients (N= 55) as well as adult HIV negative healthy donors (N= 31).
They found lower ACBP levels in EC compared to ART naïve or ART treated patients independently of age and sex. ACBP levels in EC were similar to healthy controls. ACBP levels were found to negatively correlate with CD4 count and to a lesser extent CD8 count in ART naïve patients. ACBP levels did not depend on the presence of HLA class I protective alleles and were not related to annual CD4 slope in EC.
This is an interesting cross-sectional study with original data presented as a brief report. The paper is well written. As extracellular ACBP inhibits autophagy a key element for T-cell responses, these data are of interest.
Major comment:
Although the median duration of follow-up of elite controllers was 6 years, the inclusion criteria in the group of EC (viral load < 50 in the absence of ART for at least 6 months and CD4 > 200) is unusual as 6 months is a really short period of time to characterize natural HIV control. The definition should have been more stringent (ideally > 4-5 years).
Minor comments:
- The study was probably underpowered (as noted by the authors themselves) to see the relationship between ACBP levels and weight or BMI and perhaps age as shown in other reports. Unfortunately, the different groups differ for age and sex. However, multivariate analysis confirms that the observed differences in ACBP levels were independent of sex and age.
- The authors show (Fig S1) that ACBP levels did not depend on the presence of HLA class I protective alleles. However, there is a trend to lower ACBP levels in EC with protective HLA class I and the lack of significance could be related to the low number of EC (17 with protective HLA). Can the authors comment on that ?
- There was an inverse relationship between ACBP levels and CD4 and C8 counts in ART naïve patients but not in EC. In uninfected controls, there was a strong negative relationship between ACBP and the CD4/CD8 ratio. Is there an explanation for that ?
- Regarding the relationship between ACBP levels and inflammation levels, there was no correlation for the group of EC, only a small (R= 0,29) but a significant correlation with IL-1b levels in ART treated patients. The authors discussed on inflammation levels reported to be lower in EC than in ART treated patients. However, this is not consensual (eg Noel: AIDS 2014 Feb 20;28(4):467-76) and should be rewritten accordingly.
Author Response
> We thank reviewer 1 for his valuable comments, which greatly improved our manuscript.
Major comment:
Although the median duration of follow-up of elite controllers was 6 years, the inclusion criteria in the group of EC (viral load < 50 in the absence of ART for at least 6 months and CD4 > 200) is unusual as 6 months is a really short period of time to characterize natural HIV control. The definition should have been more stringent (ideally > 4-5 years).
> We agree with reviewer 1 that duration of control is key in the characterization of EC. The only participant included less than a year after the estimated date of HIV acquisition was followed for 2.5 years in the cohort and lost control after 2 years. We have not been able to assess samples at a later timepoint than 0.7 years.
To assess the influence of time of control on ACBP levels, we compared ACBP levels in participants with less than 5, 5 to 10 or more than 10 years of HIV control. Although those controlling for 5 to 10 years had slightly higher ACBP levels than the other groups, these differences were not significant. A new figure has been added as supplementary figure S1.
Minor comments:
- The study was probably underpowered (as noted by the authors themselves) to see the relationship between ACBP levels and weight or BMI and perhaps age as shown in other reports. Unfortunately, the different groups differ for age and sex. However, multivariate analysis confirms that the observed differences in ACBP levels were independent of sex and age.
> We indeed confirm that ACBP was not linked with sex in our groups of PLWH. Also, as requested by reviewer 2, we have added in the result section that ACBP levels correlated with age only in uninfected controls.
- The authors show (Fig S1) that ACBP levels did not depend on the presence of HLA class I protective alleles. However, there is a trend to lower ACBP levels in EC with protective HLA class I and the lack of significance could be related to the low number of EC (17 with protective HLA). Can the authors comment on that ?
> Even though our study is underpowered to fully conclude on the influence of HLA, we think that lower ACBP levels in those with protective HLA are linked with stronger anti-HIV responses. We have added this hypothesis in the discussion section, that should be assessed in future studies.
- There was an inverse relationship between ACBP levels and CD4 and C8 counts in ART naïve patients but not in EC. In uninfected controls, there was a strong negative relationship between ACBP and the CD4/CD8 ratio. Is there an explanation for that ?
> CD4/CD8 ratio is considered a potent marker of immune activation and disease progression in PLWH and uninfected people. The inverse correlations between CD4, CD8 and CD4/CD8 ratio may reflect immune activation levels.
- Regarding the relationship between ACBP levels and inflammation levels, there was no correlation for the group of EC, only a small (R= 0,29) but a significant correlation with IL-1b levels in ART treated patients. The authors discussed on inflammation levels reported to be lower in EC than in ART treated patients. However, this is not consensual (eg Noel: AIDS 2014 Feb 20;28(4):467-76) and should be rewritten accordingly.
> Following Reviewer 1 suggestion, we have updated our discussion to indicate that inflammation levels in EC are not consensually lower than progressors.
Reviewer 2 Report
Isnard et al propose a descriptive study finding low levels of acyl-CoA-binding protein (ACBP) in HIV Elite Controllers when compared to HIV progressors –ART-treated or not-. As ACBP acts as a feedback inhibitor of autophagy, the authors hypothesise that the HIV antigenic presentation could explain the appropriate control of the viral infection in these patients.
The study is well constructed and clearly presented. However several improvements could be made:
Minor remarks:
- Is there any link between age and ACBP plasma levels in healthy controls?
- How many EC patients received ART previously?
- Were the EC and the primary HIV infection cohort included in the same period of time and were ACBP dosages were made at the same time in the different groups?
- Because ACBP is involved in the metabolism of lipids and benzodiazepines, it would be informative if lipolipemiants and benzodiazepine treatments were indicated if as part of the patients medication.
- We noticed that ACBP levels were not linked to HLA class I alleles. However, it would be useful to measure HLA Class I quantitative expression on PBMC (qPCR or MFI in multiparameter flow cytometry) and correlate with ACBP plasma levels. If not possible, a table with the HLA alleles and the ACBP levels could be part of main figures.
- As other inflammatory markers, ultra sensitive CRP could be measured in all groups, because is a marker of inflammation frequently studied during HIV infection.
Author Response
We thank reviewer 2 for his valuable comments which greatly improved our manuscript.
Minor remarks:
- Is there any link between age and ACBP plasma levels in healthy controls?
> Indeed, ACBP levels correlated with age only in controls (r=0.46, p<0.01). This information has been added in the results section 3.3.
- How many EC patients received ART previously?
None of our EC previously received ART.
- Were the EC and the primary HIV infection cohort included in the same period of time and were ACBP dosages were made at the same time in the different groups?
EC were collected between 2006 and 2020, ART-naïve chronically infected participants were collected between 2003 and 2012. ART-treated participants were collected between 2011 and 2016. All samples were kept at -80°C until used. Measurements were performed the same week for all groups, in batch with a mix of groups per plate. The method section has been updated to reflect this information.
- Because ACBP is involved in the metabolism of lipids and benzodiazepines, it would be informative if lipolipemiants and benzodiazepine treatments were indicated if as part of the patients medication.
> Concurrent medication information was available for EC: only 2 participants were taking rosuvastatin at the time of sample collection: ACBP 109.8 ng/mL, median of the group, and 43.8ng/mL Only 1 participant was taking diazepam at the time of sample collection: his ACBP levels were 43.8ng/mL. These information have been added to the manuscript.
We were not able to assess the influence of those medications in other groups. This limitation has been added in the discussion.
- We noticed that ACBP levels were not linked to HLA class I alleles. However, it would be useful to measure HLA Class I quantitative expression on PBMC (qPCR or MFI in multiparameter flow cytometry) and correlate with ACBP plasma levels. If not possible, a table with the HLA alleles and the ACBP levels could be part of main figures.
We agree with reviewer 2 that HLA expression is key in the establishment of effective T-cell responses. We have included in supplementary table S2 the different HLA type of our EC group and their respective ACBP levels.
- As other inflammatory markers, ultra sensitive CRP could be measured in all groups, because is a marker of inflammation frequently studied during HIV infection.
> We agree with reviewer 2 that hsCRP is an interesting inflammatory marker, however we won’t be able to perform such measurements in due time. in aging and HIV in large prospective cohorts CRP levels closely parallel IL-6 levels.